# Regional Difference in the Association between the Trajectory of Selenium Intake and Hypertension: A 20-Year Cohort Study

**DOI:** 10.3390/nu13051501

**Published:** 2021-04-29

**Authors:** Changxiao Xie, Jinli Xian, Mao Zeng, Zhengjie Cai, Shengping Li, Yong Zhao, Zumin Shi

**Affiliations:** 1Department of Nutrition and Food Hygiene, School of Public Health and Management, Chongqing Medical University, Chongqing 400016, China; 2019110926@stu.cqmu.edu.cn (C.X.); 2018111037@stu.cqmu.edu.cn (J.X.); zengmao@stu.cqmu.edu.cn (M.Z.); 2018111015@stu.cqmu.edu.cn (Z.C.); 2019110981@stu.cqmu.edu.cn (S.L.); 2Research Center for Medicine and Social Development, Chongqing Medical University, Chongqing 400016, China; 3The Innovation Center for Social Risk Governance in Health, Chongqing Medical University, Chongqing 400016, China; 4Chongqing Key Laboratory of Child Nutrition and Health, Children’s Hospital of Chongqing Medical University, Chongqing 400014, China; 5School of Public Health and Management, Chongqing Medical University, Yixueyuan Road, Yuzhong District, Chongqing 400016, China; 6Human Nutrition Department, College of Health Sciences, QU Health, Qatar University, Doha 2713, Qatar; zumin@qu.edu.qa

**Keywords:** selenium intake, hypertension, cohort study, china health, nutrition survey

## Abstract

The effect of selenium on hypertension is inconclusive. We aimed to study the relationship between selenium intake and incident hypertension. Adults (age ≥20 years) in the China Health and Nutrition Survey were followed up from 1991 to 2011 (*N* = 13,668). The latent class modeling method was used to identify trajectory groups of selenium intake. A total of 4039 respondents developed hypertension. The incidence of hypertension was 30.1, 30.5, 30.6, and 31.2 per 1000 person-years among participants with cumulative average selenium intake of 21.0 ± 5.1, 33.2 ± 2.8, 43.8 ± 3.6, and 68.3 ± 25.2 µg/day, respectively. Region and selenium intake interaction in relation to hypertension was significant. In the multivariable model, cumulative intake of selenium was only inversely associated with the incident hypertension in northern participants (low selenium zone), and not in southern participants. Compared to selenium intake trajectory Group 1 (stable low intake), all three trajectory groups had a low hazard ratio for hypertension among the northern participants. However, Group 4 (high intake and decreased) showed an increasing trend of hypertension risk in the south. In conclusion, the association between selenium intake and the incidence of hypertension varied according to regions in China. In the low soil selenium zone, high selenium intake might be beneficial for hypertension prevention.

## 1. Introduction

Hypertension is one of the most critical risk factors for cardiovascular disease and even mortality. In 2014, the prevalence of hypertension in adults (aged ≥18 years) was approximately 22% worldwide [1]. According to the China Hypertension Survey results in 2012–2015, 245 million adults (23.2% of the population) suffered from hypertension [2]. Dietary factors contribute significantly to blood pressure [3].

Selenium (Se) is an essential trace element for human health. In selenoprotein, it plays a variety of physiological functions on the body, such as antioxidant activity, prevention of viral mutations, protection of intestinal mucosa, anti-inflammatory activity, glucose metabolism effects, and insulin sensitivity effects [4]. The organic forms of selenium found in human body are selenocystein and selenoproteins. Seleproteins are known as antioxidants, which protect cells from the damage caused by free radicals. Glutathione peroxidase (GPX) is the major selenoprotein in the human body [5]. Selenium deficiency leads primarily to degeneration of many organs and tissues, and has been linked to the risk of Keshan disease, Kashin-Beck disease, cancer, cardiovascular disease and even AIDS [5,6]. Worldwide, 1/7 people are on a low-selenium diet [7]. China is among the countries heavily affected by selenium deficiency [8]. A study of residents in 10 provinces in China showed that their selenium levels decreased by 24–46% over the past two decades due to decreased grain consumption and low selenium content in rice [8]. 

Selenium affects the prevention and treatment of hypertension because of its antioxidant capacity [9,10]. Many studies showed a correlation between selenium and blood pressure. One review published in 2014 included 25 studies with more than 26,000 participants from 19 countries. Half of the 25 studies did not show any relationship between selenium and hypertension, whereas the other half found either a positive or a negative association [11]. Another systematic review published in 2019 identified six studies on the association between selenium intake and metabolic syndrome, and the association was found to be inconclusive [12]. These conflicting findings may be because selenium status is positively related to the different study populations’ soil selenium levels. Selenium has a safe narrow window. Inorganic sources of selenium exhibit higher toxicity, as compared to the organic forms [5]. A massive overdose of selenium intake may have toxic effects and cause various adverse effects, including diarrhea, hair loss, nausea, headache, nervous system disorders, and death [13]. 

Most present studies on the relationship between selenium status and hypertension used serum selenium concentration as the indicator of selenium status [11]. Few population studies have explored the relationship between selenium intake and hypertension. No longitudinal study has investigated the association between selenium intake and hypertension.

Using data from the China Health and Nutrition Survey (CHNS), we wanted to evaluate the association between selenium intake and hypertension. Furthermore, we aimed to assess the selenium intake trajectory in the general population and its association with hypertension.

## 2. Methods

### 2.1. Study Sample

The CHNS is a prospective cohort study reviewed by the University of North Carolina (USA) and the National Institute of Nutrition and Food Safety (China). This study was designed to monitor the Chinese population’s health and nutrition status using a multistage random cluster sampling method to survey nine provinces of China. Geographically, we defined Heilongjiang, Liaoning, Shandong, and Henan as the northern regions and Jiangsu, Hubei, Hunan, Guizhou, and Guangxi as the southern regions. The respondents in the cohort included all the family members of the selected family. By 2011, the CHNS had conducted nine surveys (1989, 1991, 1993, 1997, 2000, 2004, 2006, 2009, and 2011). The 1989 survey did not collect data for each age group. Thus, we only analyzed the data collected from 1991 to 2011. Due to the high loss to follow-up caused by the accelerated urbanization process, residents of new communities were invited to participate in the survey as supplementary samples since 1997. The response rate for respondents with informed consent from 1989 to 2006 was more than 60 percent. In eight surveys from 1991 to 2011, 29,220 participants aged ≥20 years old were included in the study. We excluded subjects who had no dietary intake information (*n* = 4982), those who had incredible estimates of their daily energy intake (males: >6000 kcal or <800 kcal; females: >4000 or <600 kcal) (*n* = 113), pregnant or breastfeeding females (*n* = 169), individuals with implausible BMI (Body Mass Index, <14 or >45 kg/m^2^, *n* = 7), and those who participated in only one wave (*n* = 8470). A total of 15,479 samples were used as a baseline, and their selenium intake was estimated. Focusing on newly discovered hypertension at follow-up, we excluded patients who had hypertension at baseline (*n* = 1811). 13,668 samples were finally analyzed. Those excluded in the analysis were older (44.5 vs. 39.2 years), had high BMI (23.3 vs. 22.0) and were more likely to have high income (44.8% vs. 37.5%) and live in the urban area (54.0% vs. 34.4%).

In the selenium intake trajectory analyses, we included participants who completed at least three waves of dietary intake (*N* = 11,194), including those with hypertension at baseline. After excluding those with hypertension at baseline (*n* = 1169), the remaining 10,025 participants were included in the analysis of the association between the trajectory of selenium intake and incident hypertension (Figure 1).

### 2.2. Outcome Variable: Hypertension 

This part was performed by doctors or well-trained medical workers in a community service center. Participants were told to rest for 10 min and then have their blood pressure measured while sitting. Trained staff used a mercury sphygmomanometer (well-calibrated) to measure blood pressure in participants’ right arms. Three measurements were made for each person, and the average value was considered to be the final value. For patients with diagnosed hypertension, we measured their blood pressure before taking medication. Participants were labeled as having hypertension when their average systolic blood pressure >140 mmHg and diastolic blood pressure >90 mmHg or being diagnosed with hypertension.

### 2.3. Exposure Variables: Selenium Intake

During the three-day diet survey, trained investigators weighed and recorded food and condiments from household stock, market, garden, and leftover food waste. According to China Food Composition Tables, the intake of nutrients, including selenium and protein, was collected based on the average of three-day food consumption information. Regional differences of nutrient content in food (e.g., rice) grown in different regions were reflected in the China Food Composition Table by assigning different food codes [14]. We calculated the cumulative selenium intake and used it as an exposure variable. This treatment effectively contributed to the reduced variation among individuals and represented long-term eating habits [15]. For example, if a person’s intake was x in 1991, y in 1993 and z in 1997, (x + y)/2 was the cumulative mean in 1993 and (x + y + z)/3 was the cumulative mean in 1997. The baseline intake was x and the most recent intake for 1991, 1993 and 1997 was x, y and z, respectively. Furthermore, we assigned quartiles based on all the observations.

### 2.4. Covariates 

In each survey, sociodemographic characteristics and health-related lifestyle information must be obtained through questionnaires. The socioeconomic variables were as follows: education (low, illiterate/primary school; medium, junior middle school; and high, high middle school and higher); per capita annual family income (low, medium, and high); and urbanization levels (low, medium, and high) [16]. The metabolic equivalent was estimated based on self-reported activities (including occupation, family, transportation, and recreation) and time spent and was used to reflect physical activity levels. Alcohol intake options were set as none, 1–2 times/week, 3–4 times/week, and daily. Classification variables were used to describe smoking status: non-smokers, ex-smokers, and current smokers. Overweight/obesity was defined as BMI > 24 kg/m^2^ (Chinese standards) [17,18]. Meanwhile, we calculated the prevalence of diabetes in each group.

### 2.5. Statistical Analysis 

Selenium intake was categorized into quartiles. The Chi-square test was adopted to analyze the differences of classification variables. The one-way analysis of variance was used to compare the differences of continuous variables. Cox proportional hazard model with time-varying cumulative selenium intake and covariates was used to calculate the hazard ratio (HR) for incident hypertension. Three models were established, as follows. Model 1 was adjusted for age, sex, and energy intake. Model 2 was further adjusted for fat and sodium intake, smoking, alcohol consumption, income, urbanization, education, physical activity, and BMI. Model 3 was further adjusted for two dietary patterns: the traditional southern and the modern patterns, based on a previously published factor analysis [19]. The traditional southern eating pattern is high in rice, pork, and vegetables and low in wheat. The modern diet is heavy on fruit, soy milk, eggs, milk, and fried foods. Cox proportional hazards model assumptions were examined by visual inspection of log–log plots. Multiplicity interaction between selenium intake and demographic characteristics (sex, income, education, and residence) was analyzed by adding the product of variables in the regression model. 

The latent class modeling method was applied to trajectory analysis. We constructed a group-based trajectory modeling to identify distinctive subgroups of individual trajectories of selenium intake within a population and profiling characteristics of individuals within the subgroups using the Stata command, traj. In the analysis, we included all participants who had data on their dietary intake collected at least three waves during 1991 and 2011. For each trajectory, we estimated several possible combinations of trajectory shapes (intercept only, linear, quadratic or cubic) to identify the model. The number of trajectory groups was finally determined based on two criteria: (1) the smallest value of Bayesian information criterion and (2) each group should have at least 3% of participants. Finally, the Cox proportional hazards model was adopted to analyze the relationship between selenium intake and incident hypertension risk. Model was adjusted for age, sex, intake of energy, fat and sodium, smoking, alcohol drinking, income, urban, education, physical activity, and BMI. The results were visually presented. 

STATA 16.1 was used. Statistical significance would be identified when *p* < 0.05 (two-sided).

## 3. Results

At baseline, participants in the highest quartile of cumulative selenium intake had higher income and BMI than those in the lowest quartile (Table 1). Energy and sodium intake were positively associated with selenium intake. In addition, smokers were more likely to have higher selenium intake than the others. Most of the participants had baseline selenium intake below 60 µg/day (Appendix A). The participants from the north had higher selenium intake than those from the south. At baseline before excluding those with hypertension, across quartiles of Se intake, the prevalence of hypertension was 13.2%, 12.0%, 12.3% and 13.7% in the whole sample; 11.3%, 10.4%, 11.7% and 11.5% in the south; and 18.0%, 14.4%, 13.7% and 15.6% in the north.

During a median follow-up of nine years, 4039 of 13,688 participants (for a total of 132,087 person-years) developed hypertension. Across the cumulative selenium intake quartiles, hypertension incidence was 30.1, 30.5, 30.6, and 31.2 per 1000 person-years (Table 2). In the fully adjusted model, across the quartiles of cumulative selenium intake, the HRs for incident hypertension were: 1.00, 0.91 (95% CI 0.82–1.01), 0.90 (95% CI 0.80–1.00), and 0.96 (95% CI 0.85–1.08). There was a significant positive association between the most recent selenium intake and hypertension. However, the association was attenuated and deemed statistically significant after further adjusting for dietary patterns. Baseline selenium intake was not associated with hypertension.

No interactions were observed between selenium intake and sex, income, and urbanization level to hypertension risk (Table 3). A significant selenium and region interaction was observed. In northern participants, cumulative selenium intake was inversely associated with hypertension risk. However, no such association was found in participants from the south. In contrast, high selenium intake tends to be positively associated with hypertension in participants from the south.

In group-based trajectory analysis of selenium intake between 1991 and 2011, four groups were identified (Figure 2) as follows: Group 1 (stable and low, *n* = 7149, 67.0%) had lower-than-average selenium intake profiles during all the waves; Group 2 (stable and medium, *n* = 2364, 26.3%) had a stable medium intake of Selenium; Group 3 (increasing and medium, *n* = 299, 3.7%) had a medium and increasing selenium intake; and Group 4 (descending and high, *n* = 213, 3.0%) had a high intake that decreased over time. The mean baseline selenium intakes in Groups 1, 2, 3, and 4 were 35.7, 51.2, 51.2, and 87.6 µg/day, respectively (Appendix A). There were significant differences in all sociodemographic characteristics and health-related lifestyles among the four groups. Group 4 participants were more likely to be males with high income and education than those in other groups.

Group 1 had stable low intake of Se. Group 2 had stable medium intake of Se. Group 3 had me dium Se intake and increased. Group 4 had a high intake and decreased. Dash lines represent 95% CI. Figure 3 shows the opposite associations between selenium intake trajectory groups in participants from the south and north (Group 1 was used as reference). In participants from the north, Groups 2, 3, and 4 were positively associated with incident hypertension with HRs of 0.84 (95% CI 0.74–0.95), 0.72 (95% CI 0.53–0.99), and 0.63 (95% CI 0.44–0.89), respectively. However, in participants from the south, the corresponding figures were 1.01 (95% CI 0.87–1.17), 1.34 (95% CI 0.95–1.88), and 2.09 (95% CI 1.37–3.18), respectively.

Model adjusted for age, sex, intake of energy, fat and sodium, smoking, alcohol drinking, income, urban, education, physical activity, and BMI.

Trajectory pattern was modeled using traj command in Stata. Participants had at least three waves of selenium intake. Group 1 (stable low intake) was used as the reference in multivariable Cox regression.

## 4. Discussion

In this large prospective cohort study, the selenium intake of most of the participants was below the recommended nutrient intake (60 μg/day) set by the Chinese Nutrition Society [20], and none of the participants had a selenium intake that is above the tolerable upper intake level (UL) of 400 μg/day. The cumulative average selenium intake was inversely associated with the incidence of hypertension in participants from the north but not those from the south; it was associated with overall dietary patterns. The selenium intake trajectory analysis suggested that the intake of selenium was relatively stable in the majority of the samples; only a small proportion had a slight increase in intake (3.7%) or decrease in intake (3.0%) over 20 years. This was different from the results obtained by Li et al. [8] that the overall hair selenium content of inhabitants lower 24–46% than they were 20 years ago. It was due to different research methods. We divided the selenium intake into four categories according to different selenium intake trajectories without investigating overall selenium intake.

To the best of our knowledge, this is the first longitudinal study that examines the association between selenium intake and hypertension. A cohort study conducted in Western Europe showed that low blood selenium was a risk factor for hypertension in males [21]. Western Europe is generally low in selenium, similar to the northern regions in China mentioned in this study. Two longitudinal studies in China examined the association between selenium status and incident hypertension. In a study conducted in older Chinese adults in Sichuan and Shandong Provinces, nail selenium was positively associated with incident hypertension risk [22]. However, the study excluded participants from regions with endemic diseases, including Keshan disease.

In contrast, using a sub-cohort of 2530 adults who attended the CHNS between 2009 and 2015, low serum selenium level was related to the increased risk of developing hypertension [23]. Comparing the extreme quartiles of serum selenium, the relative risk for hypertension was 1.46 (95% CI 1.11–1.93). Whether the CHNS study assessed the regional difference of the association is unknown. Studies on selenium intake and hypertension in China are limited. A cross-sectional study of 2019 adults in Hunan Province found that selenium intake was inversely associated with metabolic syndrome [24]. However, the association between selenium intake and hypertension was not statistically significant. In a case-control study conducted in Shanghai, including 550 adults, selenium intake was not associated with metabolic syndrome [25]. However, the association between selenium intake and blood pressure was not assessed.

The interaction between region and selenium intake to hypertension is interesting. Such interaction may explain the inconsistent findings on the association between selenium intake and hypertension in studies conducted outside the country. Dietary intake is the primary way for humans to consume selenium. The level of selenium is positively correlated with the protein content of the food. Selenate in plants needs to be converted into an organic form before it can be consumed by humans [26]. Selenium in foods (both animal and plant derived) is highly dependent on the amount of selenium in the environment [8,26]. Selenium soil levels are highly correlated with blood selenium levels [27], suggesting the importance of the soil environment in selenium intake. The intake of Selenium around the world varies mostly [4]. For example, the mean selenium content of soils in Enshi (China) is about 150–500 times higher than the average selenium content in selenium-deficient areas in China [28]. Geographically, China has a low selenium belt that stretches from Heilongjiang province in Northeast China to Yunnan Province in the Southeast; this area has a soil selenium content of less than 0.4 μg/g [29]. The difference in the association of selenium intake and hypertension between the north and the south may be related to the fact that the four northern provinces in our study are in the low selenium belt (Appendix A) [29]. Uncontrolled ingestion of selenium-rich products may result in poisoning [30]. Recommends for selenium vary by geographic region [26]. Our results on regional differences suggested that dietary supplement use should take the region into consideration.

Several mechanisms may explain the inverse association between selenium intake and hypertension in participants from the northern region. First, animal study results have suggested that Selenium inhibits adipocyte hypertrophy and adipogenesis [31]. In a large population study of 3214 adults in Canada, 9–27% of the change in body fat can be attributed to daily selenium intake (µg/kg/day), and the body fat percentage decreased by 3–6% with daily selenium intake increased by 1 g/kg/day [32]. In a three-month randomized trial of 27 overweight/obese individuals, selenium supplementation reinforced the effects of diet on obesity and increased lean muscle mass [33]. In China, the prevalence of obesity and hypertension is higher in the north than in the south [34,35]. Second, selenium can reduce inflammation and acquire active immunity by inhibiting the NF-kB signal [36,37]. An increasing amount of evidence suggested that immune mechanisms participate in the pathogenesis of hypertension [38].

The positive association between selenium intake and hypertension in participants from the south region is puzzling. While a similar association between serum selenium and hypertension was found in other studies in the USA [39,40], the selenium intake levels are much higher in the USA than in China. Further research is needed to validate our findings.

Our study has several strengths. First, this is a longitudinal study with multiple waves of dietary assessments using three-day food recall, combined with household food inventory records. We used cumulative mean selenium intake in our data analysis to reflect the long-term selenium intake [15]. Second, we used trajectory analysis of selenium intake to examine its relationship with hypertension. To our knowledge, this is the first study to report the relationship between the trajectory of selenium intake and hypertension risk. Furthermore, we adjusted for the overall dietary patterns using factor analysis. Third, the relatively long period of follow-up and the large sample size provides sufficient statistical power.

This study has several limitations that need to be acknowledged. First, it was unclear whether long-term intake can be adequately reflected in a three-day diet survey. The survey was not spread over four seasons. Thus, we were unable to adjust for seasonal variations in selenium intake. Second, although we adjusted for income, BMI, and dietary patterns in the multivariable analysis, residual confounding was still possible. Third, the subgroup analyses (e.g., interaction between region and trajectories of selenium intake) were explorative. Furthermore, there was no data on serum selenium.

## 5. Conclusions

In conclusion, selenium intake is relatively stable over the past two decades in the Chinese adults included in the study. Selenium intake was inversely associated with the risk of hypertension in participants from the north, but was positively associated with hypertension in participants from the south. High selenium intake appeared to be a protective factor for blood pressure in the low-selenium region. Thus, dietary supplement use should take the region into consideration.

## Figures and Tables

**Figure 1 nutrients-13-01501-f001:**
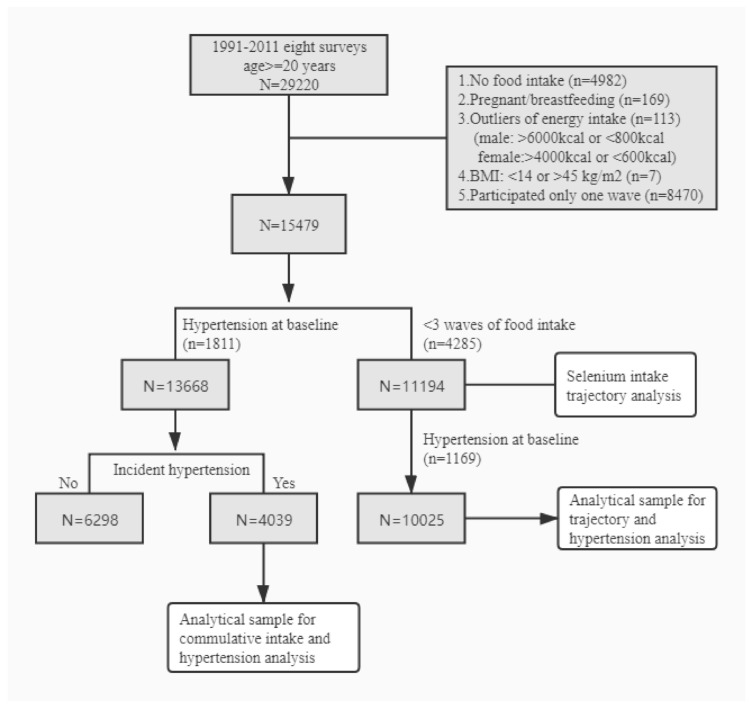
Participant flow chart.

**Figure 2 nutrients-13-01501-f002:**
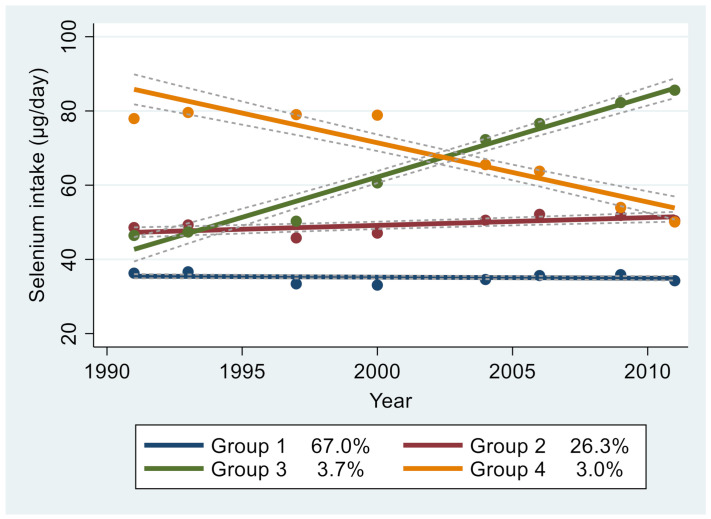
Group based trajectory of Se intake among adults attending the China Health and Nutrition Survey (CHNS), 1991–2011. (*N* = 11,194).

**Figure 3 nutrients-13-01501-f003:**
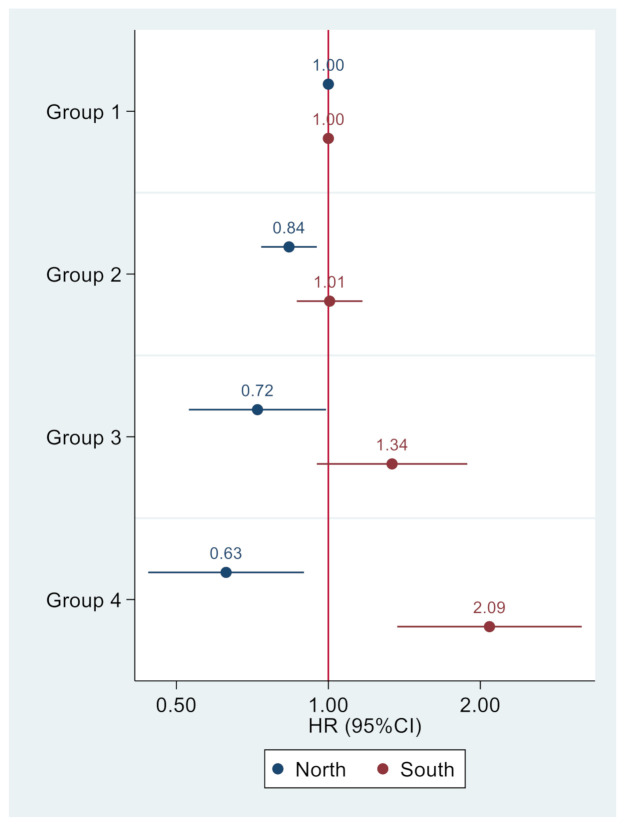
Association between trajectory of Se intake and incident hypertension among adults attending in CHNS.

**Table 1 nutrients-13-01501-t001:** Baseline sample characteristics by quartiles of Se intake: CHNS (*N* = 13,668).

Factor	Q1	Q2	Q3	Q4	*p*-Value
*N*	3453	3385	3423	3407	
Se intake (µg/day), mean (SD)	21.0 (5.1)	33.2 (2.8)	43.8 (3.6)	68.3 (25.2)	<0.001
Energy intake (kcal/day), mean (SD)	2054.1 (588.7)	2329.0 (557.9)	2572.3 (611.3)	2902.0 (759.2)	<0.001
Fat intake (g/day), mean (SD)	49.7 (29.2)	60.8 (32.8)	68.7 (35.4)	85.2 (42.5)	<0.001
Protein intake (g/day), mean (SD)	54.6 (15.5)	66.9 (15.0)	77.8 (17.4)	94.8 (24.9)	<0.001
Carbohydrate intake (g/day), mean (SD)	344.3 (120.2)	375.1 (118.6)	405.4 (137.1)	431.3 (164.2)	<0.001
Sodium intake (g/day), mean (SD)	5.7 (6.5)	5.9 (4.9)	6.4 (4.7)	7.3 (5.6)	<0.001
Traditional dietary pattern, mean (SD)	0.0 (0.8)	−0.1 (1.0)	−0.1 (1.1)	0.0 (1.4)	<0.001
Modern dietary pattern, mean (SD)	−0.6 (0.5)	−0.4 (0.7)	−0.2 (0.8)	0.3 (1.1)	<0.001
Age (years), mean (SD)	40.7 (15.6)	39.7 (14.6)	38.5 (13.5)	37.8 (13.1)	<0.001
BMI (kg/m^2^), mean (SD)	21.5 (2.8)	21.8 (2.8)	22.2 (2.9)	22.4 (3.0)	<0.001
BMI ≥24 (kg/m^2^)	17.6%	19.4%	24.2%	26.0%	<0.001
Sex					<0.001
Males	38.1%	44.3%	51.3%	61.5%	
Females	61.9%	55.7%	48.7%	38.5%	
Income					<0.001
Low	39.5%	32.0%	26.0%	21.7%	
Medium	33.3%	33.9%	33.4%	30.1%	
High	27.1%	34.1%	40.6%	48.1%	
Education					<0.001
Low	55.0%	47.7%	41.1%	35.0%	
Medium	30.5%	32.4%	33.5%	36.3%	
High	14.5%	19.9%	25.3%	28.7%	
Diabetes	4.7%	6.4%	7.2%	8.6%	0.55
Urbanization					<0.001
Low	49.4%	42.3%	38.7%	32.4%	
Medium	29.5%	29.8%	29.1%	30.3%	
High	21.1%	27.9%	32.2%	37.3%	
Smoking					<0.001
Non smoker	72.0%	68.8%	65.2%	59.0%	
Ex-smokers	1.8%	1.5%	1.6%	1.8%	
Current smokers	26.2%	29.7%	33.2%	39.2%	
Physical activity (MET-hrs/week), mean (SD)	208.8 (175.2)	211.0 (182.9)	206.4 (180.3)	191.7 (156.0)	<0.001

**Table 2 nutrients-13-01501-t002:** Hazard ratio (95% CI) for incident hypertension by quartiles of Se intake: CHNS (*N* = 4039).

			Se Intake (g/Day)		
	Q1	Q2	Q3	Q4	*p* for Trend
A. Cumulative average Se intake					
Cases	1028	1002	1005	1004	
Person-years	34,193	32,834	32,891	32,169	
Incident rate (per 1000)	30.1	30.5	30.6	31.2	
Model 1	1.00	0.98 (0.90–1.07)	1.03 (0.94–1.12)	1.19 (1.09–1.31)	<0.001
Model 2	1.00	0.92 (0.83–1.02)	0.93 (0.83–1.03)	1.01 (0.90–1.13)	0.861
Model 3	1.00	**0.91 (0.82–1.01)**	**0.90 (0.80–1.00)**	0.96 (0.85–1.08)	0.445
B. Baseline Se intake					
Cases	1056	962	1039	982	
Person-years	34,317	33,326	33,265	31,179	
Incident rate (per 1000)	30.8	28.9	31.2	31.5	
Model 1	1.00	0.97 (0.89–1.05)	1.10 (1.01–1.20)	1.15 (1.05–1.25)	<0.001
Model 2	1.00	0.93 (0.84–1.03)	1.00 (0.90–1.10)	1.00 (0.90–1.11)	0.672
Model 3	1.00	0.92 (0.83–1.01)	0.97 (0.88–1.07)	0.97 (0.87–1.07)	0.762
C. Most recent Se intake					
Cases	1050	942	1023	1024	
Person-years	33,568	33,011	33,348	32,160	
Incident rate (per 1000)	31.3	28.5	30.7	31.8	
Model 1	1.00	1.02 (0.93–1.12)	1.17 (1.07–1.28)	1.27 (1.15–1.40)	<0.001
Model 2	1.00	0.93 (0.84–1.03)	1.03 (0.93–1.15)	1.11 (0.99–1.24)	0.030
Model 3	1.00	0.92 (0.83–1.02)	1.01 (0.91–1.13)	1.08 (0.95–1.22)	0.121

Model 1 adjusted for age, sex and energy intake, Model 2 further adjusted for intake of fat and sodium, smoking, alcohol drinking, income, urban, education, physical activity and BMI. Model 3 further adjusted for dietary patterns. All the adjusted variables are treated as time-varying covariates. Bold: statistically significant.

**Table 3 nutrients-13-01501-t003:** Subgroup analyses of the association between cumulative intake and incident hypertension: CHNS (1991–2011).

	Q1	Q2	Q3	Q4	*p* for Interaction
BMI status					
Normal	1.00	0.92 (0.81–1.04)	0.88 (0.77–1.02)	1.03 (0.88–1.21)	0.496
Overweight/obese	1.00	0.88 (0.74–1.05)	0.92 (0.78–1.10)	0.89 (0.74–1.07)	
Region					
North	1.00	0.94 (0.80–1.11)	**0.82 (0.69–0.97)**	**0.84 (0.70–1.00)**	0.001
South	1.00	0.89 (0.77–1.03)	0.96 (0.82–1.13)	1.11 (0.93–1.33)	
Income					
Low	1.00	0.84 (0.71–1.00)	0.91 (0.75–1.10)	0.87 (0.69–1.09)	0.541
Medium	1.00	0.97 (0.81–1.15)	0.89 (0.73–1.07)	0.96 (0.78–1.18)	
High	1.00	0.94 (0.78–1.13)	0.94 (0.78–1.13)	1.03 (0.85–1.26)	
Sex					
Males	1.00	**0.84 (0.71–0.99)**	**0.85 (0.72–1.00)**	0.91 (0.77–1.08)	0.677
Females	1.00	0.97 (0.85–1.11)	0.94 (0.81–1.09)	0.99 (0.84–1.18)	
Urbanization level					
Low	1.00	0.83 (0.70–0.99)	0.95 (0.80–1.14)	0.94 (0.76–1.16)	0.241
Medium	1.00	0.88 (0.74–1.04)	**0.81 (0.68–0.98)**	0.98 (0.80–1.19)	
High	1.00	1.08 (0.88–1.32)	0.98 (0.80–1.21)	1.02 (0.82–1.27)	

Model adjusted for age, sex, intake of energy, fat and sodium, smoking, alcohol drinking, income, urban, education, physical activity, and BMI, and dietary patterns. Stratification variables were not adjusted in the corresponding models. Bold: statistically significant.

## Data Availability

CHNS repository. https://www.cpc.unc.edu/projects/china, accessed date: 15 December 2019.

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
