# Peer review of "Regional Difference in the Association between the Trajectory of Selenium Intake and Hypertension: A 20-Year Cohort Study"

_nutrients, 2021, doi:10.3390/nu13051501_

Round 1
Reviewer 1 Report
Xie et al examined selenium intake in relation to incidence of hypertension in a Chinese cohort study during a period of 20 years. It provides the data from a longitudinal cohort on a relationship of selenium intake with hypertension risk. Authors found that selenium intake was relatively stable over two decades in this Chinese population. High selenium intake might be beneficial for hypertension in the low soil selenium region.
I have a few comments:
1. Method: require better explanation on exposure: cumulative average Se intake, baseline Se intake and most recent Se intake plus the selenium intake trajectory analyses.
2. It would be important to present the results on group-based trajectory analysis of selenium intake in the main text instead of supplement (eg. Supplement Figure 3). Table 4 could be listed as supplement.
3. The supplement figures and table are missing for review.
Minor comments:
1. Author list in ref 7
2. Font size, bold in text through the manuscript
Author Response
Point 1: Method: require better explanation on exposure: cumulative average Se intake, baseline Se intake and most recent Se intake plus the selenium intake trajectory analyses.
Response 1: Thank you very much for your comment. We have added more detail to the manuscript. We calculated the cumulative selenium intake and used it as an exposure variable. For example, if a person’s intake was x in 1991, y in 1993 and z in 1997, (x+y)/2 was the cumulative mean in 1993 and (x+y+z)/3 was the cumulative mean in 1997. The baseline intake was x and the most recent intake for 1991, 1993 and 1997 was x, y and z, respectively.(Page 3 Line 123-126). The latent class modeling method was applied to construct trajectory groups of selenium intake. This group-based trajectory modeling method identifies distinct groups of individual trajectories of selenium intake within a population and profiling characteristics of individuals within the subgroups using the Stata command, traj (Page 4 Line 155-158). For each trajectory, we estimated several possible combinations of trajectory shapes (intercept only, linear, quadratic or cubic) to identify the model (Page 4 Line 159-161).Model was adjusted for age, sex, intake of energy, fat and sodium, smoking, alcohol drink-ing, income, urban, education, physical activity, and BMI (Page 4 Line 165-166).
Point 2: It would be important to present the results on group-based trajectory analysis of selenium intake in the main text instead of supplement (eg. Supplement Figure 3). Table 4 could be listed as supplement.
Response 2: Thank you very much for your suggestion. We have revised accordingly. The results on group-based trajectory analysis have been put in the main text. Table 4 (Supplement figure 4 actually) has been listed as supplement material. In addition, we have revised the minor comments on reference and fonts, bold, etc
Point 3: The supplement figures and table are missing for review.
Response 3: We have put all of the figures and tables in the manuscript revised.
Reviewer 2 Report
Please describe the properties of selenium as a compound with anti-oxidation properties in introduction. The language of the manuscript is not adequate for the desired journal level. The text may be handled by a native English speaker.
Abstract is not clear for the reader and do not summarize the work. For instance: no introduction to the topic. It should be reformulated. Add more details, emphasize the importance of the results. In the abstract, there is no chemical form of selenium? The authors should average this throughout the work.
Selenium is a very interesting element, but also a dangerous one. It all depends on the offer used and the chemical form. The authors use the results of the opinion publication. Add article to introduction:
Current knowledge on the importance of selenium in food for living organisms: a review. Molecules, 2016, 21(5), 609. doi:10.3390/molecules21050609.
Selenium–Fascinating Microelement, Properties and Sources in Food. Molecules, 2019, 24(7), 1298. doi:10.3390/molecules24071298
Where is the conclusion?
Discussing all these aspects will diversify the work and interest the reader.
Extend the discussion of results with the latest literature data, Discuss the results obtained with the data other authors. This will interest the reader and increase the impact of the article.
At work, I did not have a specific summary and describe the possibilities of using the results obtained in industrial practice.
Author Response
Response 1: Thank you very much for your comment. We added the following description about the antioxidant properties of selenium: The organic forms of selenium found in human body are selenocystein and selenoproteins. Seleproteins are known as antioxidants which protect cells from the damage caused by free radicals. Glutathione peroxidase (GPX) is the major selenoprotein in the human body [5] (Page 1 Line 44-47)
Point 2: Abstract is not clear for the reader and do not summarize the work. For instance: no introduction to the topic. It should be reformulated. Add more details, emphasize the importance of the results. In the abstract, there is no chemical form of selenium? The authors should average this throughout the work.
Selenium is a very interesting element, but also a dangerous one. It all depends on the offer used and the chemical form. The authors use the results of the opinion publication. Add article to introduction:
Current knowledge on the importance of selenium in food for living organisms: a review. Molecules, 2016, 21(5), 609. doi:10.3390/molecules21050609.
Selenium–Fascinating Microelement, Properties and Sources in Food. Molecules, 2019, 24(7), 1298. doi:10.3390/molecules24071298
Response 2: Thank you very much for your comment. The abstract have been revised (Page 1 Abstract). As for the importance of the results, we described it as follows: dietary supplement use should take region into consideration and emphasized it in the conclusion (Page 9 Line 285-288 & Page 11 Line 334-335). We added to the introduction a description of the toxicity of selenium and its chemical form. Inorganic sources of selenium exhibit higher toxicity as compared to the organic forms [5](Page 1 Line 63). The references you recommended are added to the manuscript as ref 5 and ref 26 respectively.
Point 3: Where is the conclusion?
Discussing all these aspects will diversify the work and interest the reader.
Extend the discussion of results with the latest literature data, Discuss the results obtained with the data other authors. This will interest the reader and increase the impact of the article.
At work, I did not have a specific summary and describe the possibilities of using the results obtained in industrial practice.
Response 3: Thank you very much for your comment. In the conclusion, we summarize the key findings of the study and the practical significance of the results (Page 11 Line 330-335). In the discussion we described more details about dietary selenium. The level of selenium is positively correlated with the protein content of the food. Selenate in plants needs to be converted into an organic form before it can be consumed by humans [26]. Selenium in foods (both animal and plant derived) is highly dependent on the amount of selenium in the environment [8,26] (Page 9 Line 273-276)
Reviewer 3 Report
Dr. Changxiao Xieand colleagues try to dissect the relationship between estimated selenium intake and the incidence of hypertension in 13,668 adults included in the China Health and Nutrition Survey followed on average for 20 years. No clear-cut association between selenium intake and the incidence of hypertension was seen but an interaction between region and selenium intake resulted significant. Indeed, only northern participants (included in a low selenium zone) but not southern participants showed the association. Compared with selenium intake trajectory, with respect to the group with a stable low intake, the other trajectory groups had a low hazard ratio for hypertension only in the northern participants. The conclusions by the authors are dubitative about the beneficial role of selenium for hypertension.
The study is of interest but I have some concerns.
- The sample size is quite large but more than half of the participants in the CNHS were excluded from the study. It should be shown what differences exist between included and excluded individuals.
- No measurement of selenium was done, even in a subgroup, rendering the estimated selenium intake highly speculative.
- Other macro or micronutrients can have had an influence on the results (how about magnesium for example?)
- It is correct to excluded people with hypertension at baseline from the analysis on the incidence of hypertension, but still they are a group of interest since they were exposed or not to selenium in the past. I would like to see a row analysis between selenium intake and the prevalence of hypertension at baseline.
- It should be clearly stated that the sub-analyses (interaction with the region, trajectories of selenium intake) are explorative.
Author Response
Response: Thank you very much for your constructive comments. We have revised the manuscript accordingly.
Point 1: The sample size is quite large but more than half of the participants in the CNHS were excluded from the study. It should be shown what differences exist between included and excluded individuals.
Response 1: Those excluded in the analysis were older (44.5 vs 39.2 years), had high BMI (23.3 vs 22.0) and were more likely to have high income (44.8% vs 37.5%) and live in urban area (54.0% vs 34.4%), We have described the difference in the method section (Page 2 Line 95-97)
Point 2: No measurement of selenium was done, even in a subgroup, rendering the estimated selenium intake highly speculative.
Response 2: Thank you for your comments. We have acknowledged the lack of data on serum selenium as a limitation. “Furthermore, there was no data on serum selenium.”(Page 11 Line 328-329).
Point 3: Other macro or micronutrients can have had an influence on the results (how about magnesium for example?)
Response 3: We agree with the opinion that other macro or micronutrients may affect hypertension. However, we have adjusted for dietary patterns in the analyses.
Point 4: It is correct to excluded people with hypertension at baseline from the analysis on the incidence of hypertension, but still they are a group of interest since they were exposed or not to selenium in the past. I would like to see a row analysis between selenium intake and the prevalence of hypertension at baseline.
Response 4: At baseline before excluding those with hypertension, across quartiles of Se intake, the prevalence of hypertension was 13.2%, 12.0%, 12.3% and 13.7% in the whole sample; 11.3%, 10.4%, 11.7% and 11.5% in the south; 18.0%, 14.4%, 13.7% and 15.6% in the north. We have described it in the results section (Page 4 Line 176-179)
Point 5: It should be clearly stated that the sub-analyses (interaction with the region, trajectories of selenium intake) are explorative.
Response 5: Thank you very much for your comment. We have commented this in the discussion (Page 11 Line 327-328) “Third, the subgroup analyses (e.g. interaction between region and trajectories of selenium intake) were explorative”.
Round 2
Reviewer 1 Report
No further comments
Reviewer 2 Report
The authors made all the corrections to the article.
Reviewer 3 Report
No further comments